# Bi-directional Task-Guided Network for Few-Shot Fine-Grained Image Classification

Zhen-Xiang Ma
Shandong University
Jinan, China
mazhenxiang0923@163.com

Zhen-Duo Chen*
Shandong University
Jinan, China
chenzd.sdu@gmail.com

Li-Jun Zhao
Shandong University
Jinan, China
lj_zhao1028@163.com

Zi-Chao Zhang
Shandong University
Jinan, China
zhangzichao1008@163.com

Tai Zheng
Shandong University
Jinan, China
zt5369623@gmail.com

Xin Luo
Shandong University
Jinan, China
luoxin.lxin@gmail.com

Xin-Shun Xu
Shandong University
Jinan, China
xuxinshun@sdu.edu.cn

## Abstract

In recent years, the Few-Shot Fine-Grained Image Classification (FS-FGIC) problem has gained widespread attention. A number of effective methods have been proposed that focus on extracting discriminative information within high-level features in a single episode/task. However, this is insufficient for addressing the cross-task challenges of FS-FGIC, which is represented in two aspects. On the one hand, from the perspective of the Fine-Grained Image Classification (FGIC) task, there is a need to supplement the model with mid-level features containing rich fine-grained information. On the other hand, from the perspective of the Few-Shot Learning (FSL) task, explicit modeling of cross-task general knowledge is required. In this paper, we propose a novel Bi-directional Task-Guided Network (BTG-Net) to tackle these issues. Specifically, from the FGIC task perspective, we design the Semantic-Guided Noise Filtering (SGNF) module to filter noise on mid-level features rich in detailed information. Further, from the FSL task perspective, the General Knowledge Prompt Modeling (GKPM) module is proposed to retain the cross-task general knowledge by utilizing the prompting mechanism, thereby enhancing the model's generalization performance on novel classes. We have conducted extensive experiments on five fine-grained benchmark datasets, and the results demonstrate that BTG-Net outperforms state-of-the-art methods comprehensively.

*Corresponding author.

## CCS Concepts

• **Computing methodologies → Visual content-based indexing and retrieval**; **Matching**.

## Keywords

Few-Shot Learning, Fine-Grained Image Analysis, Prompting Mechanism

**ACM Reference Format:**
Zhen-Xiang Ma, Zhen-Duo Chen, Li-Jun Zhao, Zi-Chao Zhang, Tai Zheng, Xin Luo, and Xin-Shun Xu. 2024. Bi-directional Task-Guided Network for Few-Shot Fine-Grained Image Classification. In *Proceedings of the 32nd ACM International Conference on Multimedia (MM '24), October 28-November 1, 2024, Melbourne, VIC, Australia.* ACM, New York, NY, USA, 10 pages. https://doi.org/10.1145/3664647.3680593

## 1 Introduction

Benefiting from the rapid advancement of deep learning [10, 14], significant progress has been achieved in the computer vision task of image classification [2, 47]. These models rely on large-scale, label-rich training samples, however, the annotation process for large-scale image datasets is extremely time-consuming and cost-consuming. In contrast, humans can effortlessly learn new concepts from one or a few samples. To mitigate the model's excessive dependence on data volume, researchers are focusing on Few-Shot Learning (FSL) [21, 29, 38].

FSL aims to use extremely limited labeled support samples based on transferable knowledge from base classes, enabling models to perform well on unlabeled query samples from novel classes. A number of successful FSL methods, based on meta-learning and metric learning, have been proposed. Inspired by this, researchers have extended FSL to more challenging fine-grained datasets, introducing the novel task of Few-Shot Fine-Grained Image Classification (FS-FGIC) [44, 54].

However, as a cross-task between Fine-Grained Image Classification (FGIC) and FSL, existing methods for FS-FGIC still have shortcomings that need further improvement, this is evident in two

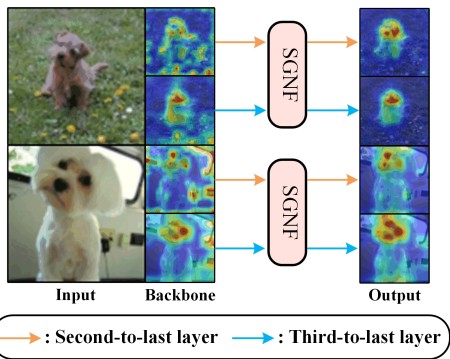

**Input**  **Backbone**  **Output**

→ : Second-to-last layer  ⇒ : Third-to-last layer

**Figure 1: Visualization results of feature maps extracted by the Conv-4 Backbone and Semantic-Guided Noise Filtering (SGNF) module on the Stanford-Dogs dataset.**

aspects. On the one hand, as a FGIC task [11], effective fine-grained feature extraction constitutes the foundation for models to achieve good performance. However, constrained by the issue of overfitting under the FSL setting, most existing methods [40, 46] primarily rely on the last layer features of the backbone for classification, lacking sufficient utilization of the network's feature learning capabilities. Specifically, the high-level features outputted by the last layer of the network often contain rich global semantic information, but this unavoidably results in the loss of local detailed information. Solely relying on high-level semantic features for classification is often insufficient to distinguish fine-grained images with subtle inter-class differences, thus impacting the performance of the method. Therefore, addressing the FS-FGIC task requires adequate assistance from local detailed features. Experience shows that mid-level features often contain more fine-grained information, but at the same time, they also include a significant amount of background and other trivial noise as shown in Figure 1. Directly using them will exacerbate overfitting issues under the FSL setting. On the other hand, as a FSL task, the essence of the model training process lies in learning the commonalities between different tasks (i.e., episodes) and based on this foundation, focusing on the unique characteristics of each task. The high degree of similarity between different classes in fine-grained data dictates the presence of abundant shared knowledge across various tasks, and the differences between tasks are mainly reflected in the details. Existing methods primarily confine their learning to within a single task, lacking explicit modeling of cross-task general knowledge. Clearly, under the FSL setting, especially for fine-grained datasets, general knowledge facilitates transfer learning between tasks. If the model only focuses on intra-task feature extraction, it will cause the model to overfit on the trainable base classes, preventing effective learning of task-transferable knowledge for generalization to unseen novel classes, thus affecting the performance of FS-FGIC.

To address the cross-task challenge of FS-FGIC, we propose a novel Bi-directional Task-Guided Network (BTG-Net), which contains two modules to solve the feature requirements of FS-FGIC from the perspective of two tasks respectively. From the perspective of the FGIC task, it is essential to introduce rich local detail features to assist classification, while reducing the overfitting issue as much

as possible. Specifically, instead of simply concatenating high-level and mid-level features, we propose the Semantic-Guided Noise Filtering (SGNF) module to generate filters for mid-level features through high-level features with powerful target object localization capability. These filters are then employed to re-weight and adjust the mid-level features, effectively eliminating background and other trivial noises as shown in Figure 1. This process provides the model with improved fine-grained information to assist in classification. From the perspective of the FSL task, it is necessary to learn cross-task general knowledge. We design the General Knowledge Prompt Modeling (GKPM) module, which utilizes high-level features to learn cross-task general knowledge, as these semantically rich high-level features often represent general attributes of categories. Inspired by the prompting mechanism in computer vision [18], a learnable Task-General Prompt is designed to learn shared general knowledge across all tasks. By introducing cross-task general knowledge and using the self-attention mechanism of Transformer [9], the query features are further optimized to reduce the distance between query features and prototypes of the same class while distancing them from prototypes of different classes, with limited support samples. This process enables the model to maintain robust generalization capabilities on the novel classes.

In general, BTG-Net is dedicated to meeting the feature needs of both fine-grained task and few-shot learning task, and thereby achieving state-of-the-art results on five fine-grained benchmarks. Our contributions can be summarized as follows:

- We propose a novel BTG-Net that addresses the feature requirements of the FS-FGIC from the perspectives of both FGIC and FSL tasks.
- From the perspective of the FGIC task, we propose the SGNF module, which effectively filters out noise such as background in mid-level features with the assistance of semantic guidance of high-level features. This process introduces rich fine-grained information to assist FS-FGIC, while minimizing overfitting problem associated with the addition of features as much as possible.
- From the perspective of the FSL task, we propose the GKPM module. We design the Task-General Prompt to retain the cross-task general knowledge, enabling the model to exhibit strong generalization capabilities on novel tasks and classes.
- We conducted extensive experiments on five few-shot fine-grained benchmark datasets, and the results indicate that the proposed method outperforms the current state-of-the-art methods comprehensively.

## 2 Related Work

### 2.1 Few-Shot Learning

FSL aims to recognize novel classes with only a limited number of support samples provided for each class. Previous research commonly utilizes a meta-learning framework, where a model is trained on a series of few-shot training tasks/episodes randomly sampled from the base dataset to rapidly adapt to unseen testing tasks. Specifically, meta-learning based methods [22, 30] try to discover a suitable gradient-based optimization strategy to quickly adapt to novel tasks with few gradient updates. MAML [12] aims to learn a way to initialize parameters so that the model can easily adapt to new tasks.

Data-augmentation based methods [3, 13, 35] seek to train a generator from base classes and leverage it to generate novel samples. Metric-learning based methods [33, 34, 38] focus on learning a fixed embedding function to map input images into a low-dimensional embedding space. FRN [43] uses support sample features to reconstruct query sample features for better similarity measurement.

However, traditional FSL methods, which usually pay less attention to feature learning, may not always perform well on fine-grained datasets with subtle inter-class differences. Therefore, FS-FGIC has attracted increasing attention in recent years.

## 2.2 Few-Shot Fine-Grained Image Classification

Expanding general FSL methods directly to fine-grained datasets does not yield satisfactory performance, therefore, Wei et al. [41] first define the FS-FGIC task and use bilinear features to train a classifier with piecewise mapping. BSNet [26] utilizes two metric functions for the joint optimization of the model. PaCL [40] employs contrastive learning to extract more discriminative features from various object regions. TDM [23] generates task-specific channel weights to localize the discriminative regions for each class. BiFRN [44] introduces a bi-reconstruction mechanism to simultaneously accommodate for inter-class and intra-class variations.

The existing method AGPF [36] attempts to aggregate multi-scale information from different layers, but it neglects that introducing mid-level features with background noise exacerbates the overfitting issue under the FSL setting. BSFA [49] proposes a two-stage background suppression and foreground alignment framework to capture subtle differences and align support and query samples. This is an operation within high-level features, neglecting the rich detailed information in mid-level features. Additionally, BFSA directly crops the background of high-level features, not simply eliminating non-generalizable background noise. In contrast, we use high-level features to filter out background noise in mid-level features, thereby providing fine-grained information to assist classification in the FS-FGIC task. Furthermore, all the mentioned methods are confined to feature learning within a single task, overlooking the necessity for exploring commonalities across FSL tasks. In contrast, our proposed method explicitly models cross-task general knowledge using the prompting mechanism, enhancing the model's generalization capability on novel classes.

## 2.3 Prompting Mechanism

Inspired by prompting mechanism in the field of natural language processing [1], VPT [18] is the first to apply prompting mechanism to computer visual tasks. Existing prompt-based methods [52, 53] typically require freezing large-scale pre-trained models, designing learnable prompts for parameter-efficient fine-tuning to adapt to downstream tasks. In contrast, our proposed BTG-Net does not require pre-trained models nor parameter freezing; the entire model is trained end-to-end.

## 3 Method

### 3.1 Problem Definition

Under the standard FSL settings, the dataset $D = \{(x_i, y_i), y_i \in Y)\}$ is divided into the training dataset $D_{base} = \{(x_i, y_i), y_i \in Y_{base})\}$, the validation dataset $D_{val} = \{(x_i, y_i), y_i \in Y_{val})\}$, and the test

dataset $D_{novel} = \{(x_i, y_i), y_i \in Y_{novel})\}$, where $x_i$ is the original sample, $y_i$ is the $i$-th class label, and $Y_{base} \cap Y_{val} \cap Y_{novel} = \emptyset$. We adopt the episodic training mechanism. Specifically, during the training phase, each episode represents an $N$-way $K$-shot classification problem. $N$ classes are randomly selected from $D_{base}$, each class having $K$ labeled images in the support set and $U$ unlabeled images in the query set. During the testing phase, the model is evaluated on the test dataset $D_{novel}$. Without loss of generality, the following discussion assumes consideration within an episode/task.

### 3.2 Overall Framework

To address the cross-task problem of FS-FGIC from the perspectives of both FGIC and FSL tasks, we propose the novel BTG-Net, and the overall framework is shown in Figure 2.

The BTG-Net consists of four components: The first component is the backbone network, which is used to extract deep features from the support and query samples. The second component is the SGNF Module, which takes the high-level features (output from the last layer of the backbone network) and mid-level features (from the second-to-last and third-to-last layers) as inputs. It utilizes the high-level features to perform noise filtering on the mid-level features. The third component is the GKPM module, which takes high-level features as input and utilizes the prompting mechanism to model task-general knowledge, so as to adjust query features and enhance the generalization ability of the model. The last component is the Similarity module, responsible for computing the cosine similarity between the query features and prototypes at each layer. The weighted sum of the three similarities is employed for the classification of the query samples.

### 3.3 Semantic-Guided Noise Filtering

Through the analysis from the perspective of the FGIC task in the introduction section, globally semantic-rich high-level features (i.e., the output of the network's last layer) cannot effectively handle fine-grained images with subtle differences between categories. Simply concatenating mid-level features will introduce trivial noise such as background, exacerbating the overfitting issue under the FSL setting. Therefore, we designed the Semantic-Guided Noise Filtering (SGNF) module, as illustrated in Figure 2, to achieve a filtering effect of high-level features on mid-level features.

Specifically, given a sample $\mathbf{X}$, we obtain features through the backbone network $f$ as follows,

$$\mathbf{F}_l, \mathbf{F}_s, \mathbf{F}_t = f(\mathbf{X}), \qquad (1)$$

where $\mathbf{F}_l \in \mathbb{R}^{C^l \times H^l \times W^l}$, $\mathbf{F}_s \in \mathbb{R}^{C^s \times H^s \times W^s}$ and $\mathbf{F}_t \in \mathbb{R}^{C^t \times H^t \times W^t}$ represent the feature maps obtained from the last layer, second-to-last layer, and third-to-last layer, respectively.

Due to the rich global semantic information of the high-level feature $\mathbf{F}_l$, which can effectively locate the global information of the target, we use the $\mathbf{F}_l$ to generate filters for mid-level features $\mathbf{F}_s$ and $\mathbf{F}_t$. Considering the varying sizes of feature maps at different layers, we use Bilinear interpolation $B(\cdot)$ to map the size of the feature map of $\mathbf{F}_l$ to match that of $\mathbf{F}_s$ and $\mathbf{F}_t$. It is important to note that the use of bilinear interpolation does not alter the number of channels in the feature maps, thereby avoiding any loss of information. The

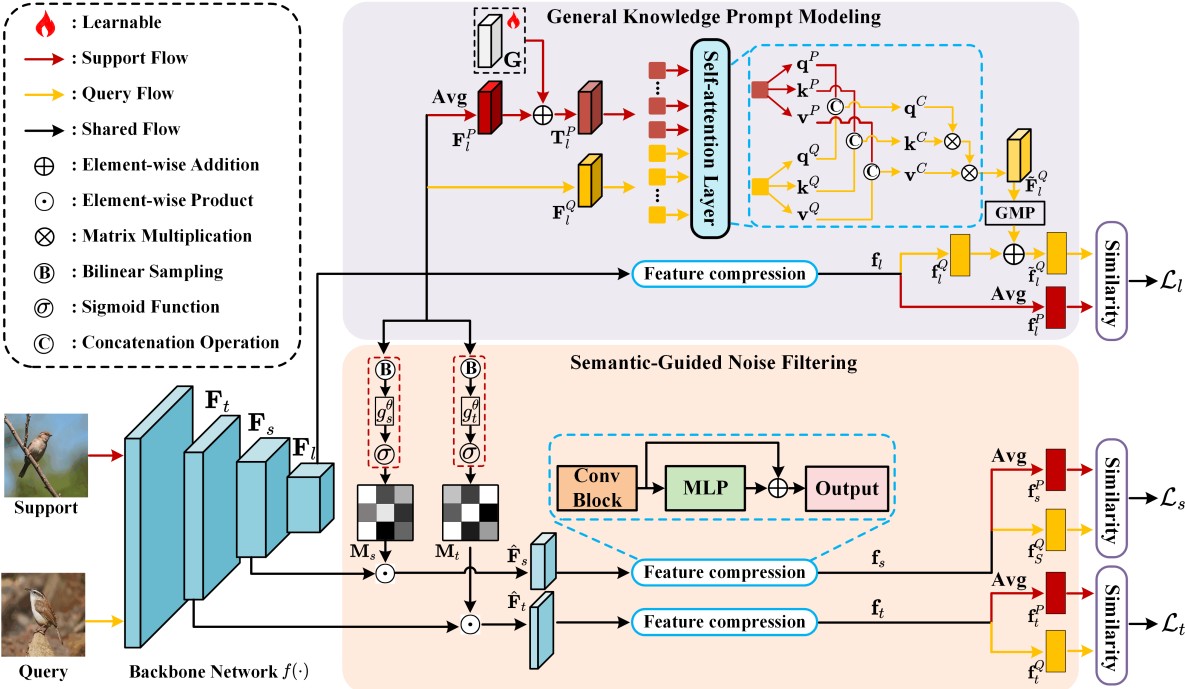

**Figure 2: Overview of the Bi-directional Task-Guided Network (BTG-Net), consisting of the Backbone Network, the Semantic-Guided Noise Filtering (SGNF) module, the General Knowledge Prompt Modeling (GKPM) module, and the Similarity module.**

filters can be formulated as

$$\mathbf{M}_s = \sigma(g_s^\theta(B(\mathbf{F}_l))) \in \mathbb{R}^{1 \times H^s \times W^s}, \tag{2}$$

$$\mathbf{M}_t = \sigma(g_t^\theta(B(\mathbf{F}_l))) \in \mathbb{R}^{1 \times H^t \times W^t}, \tag{3}$$

where $g_s^\theta$ and $g_t^\theta$ are sub-networks that include two convolutional layers each, and $\sigma$ is the sigmoid activation function.

Based on $\mathbf{M}_s$ and $\mathbf{M}_t$, we can reweight and adjust the mid-level features to eliminate trivial noise such as background, which can be formulated as

$$\hat{\mathbf{F}}_s = \mathbf{F}_s \odot \mathbf{M}_s \in \mathbb{R}^{C^s \times H^s \times W^s}, \tag{4}$$

$$\hat{\mathbf{F}}_t = \mathbf{F}_t \odot \mathbf{M}_t \in \mathbb{R}^{C^t \times H^t \times W^t}, \tag{5}$$

where $\odot$ denotes element-wise product operation with the broadcasting mechanism. Next, we design the feature compression block to compress the feature spaces of $\hat{\mathbf{F}}_s$ and $\hat{\mathbf{F}}_t$, mitigating potential overfitting issues. Features $\mathbf{f}_s \in \mathbb{R}^{C^s}$ and $\mathbf{f}_t \in \mathbb{R}^{C^t}$ can be obtained as follows,

$$\mathbf{f}_s = \alpha(Conv_s(\hat{\mathbf{F}}_s)) + (1-\alpha)(MLP_s(Conv_s(\hat{\mathbf{F}}_s))), \tag{6}$$

$$\mathbf{f}_t = \alpha(Conv_t(\hat{\mathbf{F}}_t)) + (1-\alpha)(MLP_t(Conv_t(\hat{\mathbf{F}}_t))), \tag{7}$$

where $Conv_s(\cdot)$ and $Conv_t(\cdot)$ each represents convolution block with global max pooling, $MLP_s(\cdot)$ and $MLP_t(\cdot)$ each represents fully connected block, and $\alpha$ is a balancing hyper-parameter. Finally, query features $(\mathbf{f}_s^Q, \mathbf{f}_t^Q)$ and support features $(\mathbf{f}_s^S, \mathbf{f}_t^S)$ can be separated from $\mathbf{f}_s$ and $\mathbf{f}_t$, respectively. Following the ProtoNet [33],

we generate prototypes $(\mathbf{f}_s^P, \mathbf{f}_t^P)$ for each class. The computation for the prototype for the $i$-th class is as follows,

$$\mathbf{f}_{s,i}^P = \frac{1}{K} \sum_{j=1}^{K} \mathbf{f}_{i,j,s}^S, \tag{8}$$

$$\mathbf{f}_{t,i}^P = \frac{1}{K} \sum_{j=1}^{K} \mathbf{f}_{i,j,t}^S. \tag{9}$$

In this module, we extensively explored the network's feature extraction capabilities. By filtering and eliminating noise from mid-level features, we provide rich local details to assist in classification for the FS-FGIC task. This procedure also alleviates overfitting issues under the FSL setting.

### 3.4 General Knowledge Prompt Modeling

After addressing the demand for fine-grained features from the FGIC task perspective, we further analyzed the FSL task. If the model is confined to feature learning within a single task, it can not explicitly learn transferable knowledge, resulting in overfitting on the trainable base classes and poor generalization performance on novel classes. In light of this, we designed the General Knowledge Prompt Modeling (GKPM) module as illustrated in Figure 2, employing the prompting mechanism to explicitly model cross-task general knowledge, and then introducing it into query features to ensure the model possesses robust generalization capabilities on novel classes.

**Table 1: The splits of datasets, $C_{train}$, $C_{val}$, $C_{test}$ represent the number of classes for training, validation, and test, respectively. $C_{total}$ indicates the original number of classes.**

| Dataset | $C_{train}$ | $C_{val}$ | $C_{test}$ | $C_{total}$ |
|---|---|---|---|---|
| CUB-200-2011 | 100 | 50 | 50 | 200 |
| Stanford-Dogs | 70 | 20 | 30 | 120 |
| Stanford-Cars | 130 | 17 | 49 | 196 |
| meta-iNat | 908 | - | 227 | 1135 |
| tiered meta-iNat | 781 | - | 354 | 1135 |

Specifically, to express clarity, we use $\mathbf{F}_l^S \in \mathbb{R}^{C^l \times H^l \times W^l}$ and $\mathbf{F}_l^Q \in \mathbb{R}^{C^l \times H^l \times W^l}$ to represent the support and query features, respectively. Then we generate prototypes for each class as follows,

$$\mathbf{F}_{l,i}^P = \frac{1}{K} \sum_{j=1}^{K} \mathbf{F}_{i,j,l}^S. \tag{10}$$

Thereafter, inspired by the prompting mechanism, we design the learnable Task-General (TG) Prompt ($\mathbf{G} \in \mathbb{R}^{C^l \times H^l \times W^l}$) to retain cross-task general knowledge. It is important to note that the TG Prompt is shared across all tasks during the training, validation, and testing phases. To incorporate cross-task general knowledge into the specific task, we add $\mathbf{G}$ with each prototype to obtain the task generalization information, which is formulated as

$$\mathbf{T}_l^P = \mathbf{F}_l^P + \mathbf{G} \in \mathbb{R}^{C^l \times H^l \times W^l}, \tag{11}$$

we omit the subscript $i$ for clarity in expression. Based on $\mathbf{T}_l^P$, we dynamically adjust the query features $\mathbf{F}_l^Q$ using the self-attention mechanism of the Transformer. Specifically, we flatten $\mathbf{F}_l^Q$ and $\mathbf{T}_l^P$ into $(H^l \times W^l)$ tokens, each token is $C^l$-dimensional. Then, we use weighted parameters $W_{qkv}$ to map each token to the query vector $\mathbf{q}$, key vector $\mathbf{k}$, and value vector $\mathbf{v}$, as follows,

$$\mathbf{q}^Q, \mathbf{k}^Q, \mathbf{v}^Q = W_{qkv}^Q \mathbf{F}_l^Q \in \mathbb{R}^{H^l W^l \times C^l}, \tag{12}$$

$$\mathbf{q}^P, \mathbf{k}^P, \mathbf{v}^P = W_{qkv}^P \mathbf{T}_l^P \in \mathbb{R}^{H^l W^l \times C^l}. \tag{13}$$

Afterward, to adequately model the relationship between query features $\mathbf{F}_l^Q$ and the task generalization information $\mathbf{T}_l^P$, we concatenate vectors of the same type as follows,

$$\begin{cases} \mathbf{q}^C = \begin{bmatrix} \mathbf{q}^Q, \mathbf{q}^P \end{bmatrix} \\ \mathbf{k}^C = \begin{bmatrix} \mathbf{k}^Q, \mathbf{k}^P \end{bmatrix} \\ \mathbf{v}^C = \begin{bmatrix} \mathbf{v}^Q, \mathbf{v}^P \end{bmatrix} \end{cases} \in \mathbb{R}^{2H^l W^l \times C^l}, \tag{14}$$

where $[\cdot, \cdot]$ is the concatenation operation, then we use standard self-attention mechanism in Transformer to obtain the task generalization query features $\hat{\mathbf{F}}_l^Q$. This process can be represented as

$$\tilde{\mathbf{F}}_l^Q = \text{softmax}\left( \frac{\mathbf{q}^C (\mathbf{k}^C)^T}{(C^l)^{\frac{1}{4}}} \right) \mathbf{v}^C \in \mathbb{R}^{2H^l W^l \times C^l}. \tag{15}$$

To maintain task generalization query features with the same feature map size as the original query features, we slice $\tilde{\mathbf{F}}_l^Q$, retaining only the first $HW$ vectors, and then reshape it back to the size of $C^l \times H^l \times W^l$, i.e. $\tilde{\mathbf{F}}_l^Q \in \mathbb{R}^{C^l \times H^l \times W^l}$.

**Table 2: The accuracy (%) of 5-way 1-shot and 5-shot on CUB (using raw images) dataset. The best results are shown in bold.**

| Methods | Backbone | CUB 1-shot | CUB 5-shot |
|---|---|---|---|
| ProtoNet (NeurIPS-17) [33] | Conv-4 | 59.95 | 76.01 |
| MattML (IJCAI-20) [54] | Conv-4 | 66.29 | 80.34 |
| PoseNorm (CVPR-20) [37] | Conv-4 | 64.17 | 81.96 |
| FEAT (CVPR-20) [48] | Conv-4 | 68.87 | 82.90 |
| BSNet (TIP-21) [26] | Conv-4 | 55.81 | 76.34 |
| FRN (CVPR-21) [43] | Conv-4 | 69.45 | 85.16 |
| OLSA (MM-21) [45] | Conv-4 | 73.07 | 86.24 |
| ATT-NET (AAAI-22) [46] | Conv-4 | 72.89 | 86.60 |
| FRN + TDM (CVPR-22) [23] | Conv-4 | 71.37 | 86.45 |
| AGPF (PR-22) [36] | Conv-4 | 74.03 | 86.54 |
| PaCL (MM-22) [40] | Conv-4 | 74.04 | 88.75 |
| BiFRN (AAAI-23) [44] | Conv-4 | 74.98 | 89.14 |
| Ours | Conv-4 | **79.19** | **89.71** |
| ProtoNet (NeurIPS-17) [33] | ResNet-12 | 72.99 | 86.65 |
| Neg-Cosine (ECCV-20) [27] | ResNet-18 | 72.66 | 89.40 |
| P-Transfer (AAAI-21) [31] | ResNet-12 | 73.16 | 88.32 |
| DeepEMD (CVPR-20) [51] | ResNet-12 | 75.65 | 88.69 |
| OLSA (MM-21) [45] | ResNet-12 | 77.77 | 89.87 |
| FRN (CVPR-21) [43] | ResNet-12 | 83.55 | 92.92 |
| ISC (TIP-22) [5] | ResNet-12 | 73.72 | 87.51 |
| AGPF (PR-22) [36] | ResNet-12 | 78.73 | 89.77 |
| PaCL (MM-22) [40] | ResNet-12 | 77.80 | 92.07 |
| FRN + TDM (CVPR-22) [23] | ResNet-12 | 84.36 | 93.37 |
| BSFA (TSCVT-23) [49] | ResNet-12 | 82.27 | 90.76 |
| FGFL (ICCV-23) [4] | ResNet-12 | 80.77 | 92.01 |
| Ventral (AAAI-23) [8] | ResNet-12 | 83.33 | 92.97 |
| BiFRN (AAAI-23) [44] | ResNet-12 | 82.07 | 92.11 |
| Ours | ResNet-12 | **86.44** | **94.20** |

The original feature $\mathbf{F}_l$ compresses the feature space in the same way as the previous module, as

$$\mathbf{f}_l = \alpha(Conv_l(\mathbf{F}_l)) + (1 - \alpha)(MLP_l(Conv_l(\mathbf{F}_l))). \tag{16}$$

Based on $\mathbf{f}_l \in \mathbb{R}^{C^l}$, we can obtain query features $\mathbf{f}_l^Q$ and support features $\mathbf{f}_l^S$, and then acquire prototype $\mathbf{f}_l^P$ using the same approach as in Eq (8). Finally, we integrate task generalization query features $\tilde{\mathbf{F}}_l^Q$ into the query features to obtain a comprehensive representation that includes cross-task information, as follows,

$$\tilde{\mathbf{f}}_l^Q = \mathbf{f}_l^Q + \lambda GMP(\tilde{\mathbf{F}}_l^Q), \tag{17}$$

where $\lambda$ is the learnable weight parameter and $GMP(\cdot)$ is Global Max Pooling.

Through the above procedure, we store cross-task general knowledge with the TG prompt and introduce it to optimize the query features, bringing the query closer to the prototype of the same class, further enhancing intra-class consistency feature learning with limited support samples. This aids in the FS-FGIC learning of well-transferable meta-knowledge, enabling the model to exhibit

**Table 3: The accuracy (%) of 5-way 1-shot and 5-shot on Stanford-Dogs and Stanford-Cars datasets.**

| Methods | Backbone | Stanford-Dogs | | Stanford-Cars | |
|---|---|---|---|---|---|
| | | 1-shot | 5-shot | 1-shot | 5-shot |
| ProtoNet (NeurIPS-17) [33] | Conv-4 | 40.81 ± 0.83 | 61.58 ± 0.71 | 36.51 ± 0.74 | 62.14 ± 0.76 |
| MattML (IJCAI-20) [54] | Conv-4 | 54.84 ± 0.53 | 71.34 ± 0.38 | 66.11 ± 0.54 | 82.80 ± 0.28 |
| ATL-Net (IJCAI-20) [7] | Conv-4 | 54.49 ± 0.92 | 73.20 ± 0.69 | 67.95 ± 0.84 | 89.16 ± 0.48 |
| BSNet (TIP-21) [26] | Conv-4 | 43.13 ± 0.85 | 62.61 ± 0.73 | 44.56 ± 0.83 | 63.72 ± 0.78 |
| LRPABN (TMM-21) [17] | Conv-4 | 45.72 ± 0.75 | 60.94 ± 0.66 | 60.28 ± 0.76 | 73.29 ± 0.58 |
| FRN (CVPR-21) [43] | Conv-4 | 49.37 ± 0.20 | 67.13 ± 0.17 | 58.90 ± 0.22 | 79.65 ± 0.15 |
| TOAN (TCSVT-21) [16] | Conv-4 | 49.30 ± 0.77 | 67.16 ± 0.49 | 65.90 ± 0.72 | 84.24 ± 0.48 |
| OLSA (MM-21) [45] | Conv-4 | 55.53 ± 0.45 | 71.68 ± 0.36 | 70.13 ± 0.48 | 84.29 ± 0.31 |
| ATT-NET (AAAI-22) [46] | Conv-4 | 59.81 ± 0.50 | 77.19 ± 0.35 | 70.21 ± 0.50 | 85.55 ± 0.31 |
| AGPF (PR-22) [36] | Conv-4 | 60.89 ± 0.98 | 78.14 ± 0.62 | 78.14 ± 0.84 | 87.42 ± 0.57 |
| PaCL (MM-22) [40] | Conv-4 | 59.76 ± 0.70 | 77.50 ± 0.48 | 72.21 ± 0.68 | 88.02 ± 0.36 |
| HelixFormer (MM-22) [50] | Conv-4 | 59.81 ± 0.50 | 73.40 ± 0.36 | 75.46 ± 0.37 | 89.68 ± 0.25 |
| LCCRN (TCSVT-23) [25] | Conv-4 | - | - | 71.62 ± 0.21 | 86.41 ± 0.12 |
| BiFRN (AAAI-23) [44] | Conv-4 | 61.39 ± 0.23 | 78.86 ± 0.15 | 76.22 ± 0.20 | 90.66 ± 0.11 |
| Ours | Conv-4 | **67.33 ± 0.52** | **81.08 ± 0.33** | **78.72 ± 0.46** | **90.78 ± 0.26** |
| BSNet (TIP-21) [26] | ResNet-12 | 61.95 ± 0.97 | 79.62 ± 0.63 | 71.07 ± 1.03 | 88.38 ± 0.62 |
| OLSA (MM-21) [45] | ResNet-12 | 64.15 ± 0.49 | 78.28 ± 0.32 | 77.03 ± 0.46 | 88.85 ± 0.46 |
| AGPF (PR-22) [36] | ResNet-12 | 72.34 ± 0.86 | 84.02 ± 0.57 | 85.34 ± 0.74 | 94.79 ± 0.35 |
| HelixFormer (MM-22) [50] | ResNet-12 | 65.92 ± 0.49 | 80.65 ± 0.36 | 79.40 ± 0.43 | 92.26 ± 0.15 |
| BSFA (TSCVT-23) [49] | ResNet-12 | 69.58 ± 0.50 | 82.59 ± 0.33 | 88.93 ± 0.38 | 95.20 ± 0.20 |
| LCCRN (TCSVT-23) [25] | ResNet-12 | - | - | 87.04 ± 0.17 | 96.09 ± 0.07 |
| BiFRN (AAAI-23) [44] | ResNet-12 | 72.54 ± 0.22 | 85.86 ± 0.13 | 88.43 ± 0.17 | 96.34 ± 0.07 |
| Ours | ResNet-12 | **76.06 ± 0.50** | **88.48 ± 0.29** | **90.28 ± 0.34** | **96.78 ± 0.15** |

**Table 4: The accuracy (%) of 5-way 1-shot and 5-shot on meta-iNat and tiered meta-iNat datasets using Conv-4 backbone.**

| Methods | meta-iNat | | tiered meta-iNat | |
|---|---|---|---|---|
| | 1-shot | 5-shot | 1-shot | 5-shot |
| ProtoNet (NeurIPS-17) [33] | 53.78 | 73.80 | 35.47 | 54.85 |
| DN4 (CVPR-19) [24] | 62.32 | 79.76 | 43.82 | 64.17 |
| DSN (CVPR-20) [32] | 58.08 | 77.38 | 36.82 | 60.11 |
| CTX (NeurIPS-20) [6] | 60.03 | 78.80 | 36.83 | 60.84 |
| DeepEMD (CVPR-20) [51] | 54.48 | 68.36 | 36.05 | 48.55 |
| FRN (CVPR-21) [43] | 61.98 | 80.04 | 43.95 | 63.45 |
| FRN + TDM (CVPR-22) [23] | 63.97 | 81.60 | 44.05 | 62.91 |
| MCL (CVPR-22) [28] | 64.66 | 81.31 | 44.08 | 64.61 |
| MCL-Katz (CVPR-22) [28] | 63.92 | 81.09 | 44.00 | 64.24 |
| BiFRN (AAAI-23) [44] | 66.07 | 83.30 | 46.64 | 66.46 |
| Ours | **71.36** | **84.59** | **50.62** | **69.11** |

robust generalization capabilities on novel classes. Experimental results also validate this claim.

### 3.5 Overall Objectives

After the SGNF module and GKPM module, we obtained the outputs of three layers: $(\tilde{\mathbf{f}}_l^Q, \mathbf{f}_l^P)$, $(\mathbf{f}_s^Q, \mathbf{f}_s^P)$, $(\mathbf{f}_t^Q, \mathbf{f}_t^P)$, we simplify the symbols

to achieve a clearer representation, i.e. $(\mathbf{f}_m^Q, \mathbf{f}_m^P)$, where $m = \{l, s, t\}$, with $\tilde{\mathbf{f}}_l^Q$ simplified to $\mathbf{f}_l^Q$.

In a $N$-way $K$-shot training episodic, the loss function can be formulated as follows,

$$\mathcal{L}_m = -\frac{1}{N}\frac{1}{U}\sum_{i=1}^{N}\sum_{j=1}^{U} log \frac{\exp\left(\tau_m \mathcal{S}\left(\mathbf{f}_{i,j,m}^Q, \mathbf{f}_{i,m}^P\right)\right)}{\sum_{p=1}^{N} \exp\left(\tau_m \mathcal{S}\left(\mathbf{f}_{i,j,m}^Q, \mathbf{f}_{p,m}^P\right)\right)}, \quad (18)$$

where $\tau_m$ are learnable scaling parameters and $\mathcal{S}(\cdot)$ represents the Cosine similarity to measure the similarity between query features and the prototype. The overall loss is calculated as

$$\mathcal{L} = \mathcal{L}_l + \mathcal{L}_s + \mathcal{L}_t. \quad (19)$$

## 4 Experiments

### 4.1 Dataset

We use five widely-used few-shot fine-grained benchmark datasets, including **CUB-200-2011** [39], **Stanford Dogs** [19], **Stanford Cars** [20], **meta-iNat** [15, 42], and **tiered meta-iNat** [42]. We follow the commonly used standard dataset splitting way [43, 54], as shown in Tab. 1.

### 4.2 Implementation Details

For a fair comparison with previous work, we adopt two commonly used Conv-4 and ResNet-12 as the backbone networks, and both accepted input image sizes of $84 \times 84$.

**Table 5: Ablation study of submodules using Conv-4 backbone on the CUB and Stanford-Dogs datasets.**

| Baseline | SGNF | GKPM | CUB | | Stanford-Dogs | |
|---|---|---|---|---|---|---|
| | | | 1-shot | 5-shot | 1-shot | 5-shot |
| ✓ | | | 68.03 | 81.92 | 55.85 | 71.35 |
| ✓ | ✓ | | 75.08 | 87.03 | 60.64 | 75.98 |
| ✓ | | ✓ | 72.68 | 85.32 | 60.29 | 74.82 |
| ✓ | ✓ | ✓ | **79.19** | **89.71** | **67.33** | **81.08** |

**Meta-training Details:** We employ common data augmentation techniques, such as random crop, horizontal flip, and color jitter, consistent with existing methods. For all benchmark datasets, we use SGD with Nesterov momentum of 0.9. The initial learning rate is set to 0.1 and weight decay to 5e-4. The $\alpha$ is set to 0.3.

**Meta-testing Details:** We apply the standard 5-way 1-shot and 5-way 5-shot settings. We save the best-performing model on the validation set to test. We report the 95% confidence interval results for 2,000 test episodes for all experiments.

### 4.3 Comparison with State-of-the-Art Methods

In this section, we compare our proposed method with a number of state-of-the-art methods, including general FSL methods (e.g. ProtoNet, DeepEMD, and FRN) and FS-FGIC methods (e.g. Helix-Former, AGPF, and BiFRN). The experimental results are shown in Table 2-4. Based on the experimental results, it can be observed:

Firstly, our proposed method outperforms state-of-the-art methods, achieving comprehensive performance leadership. This superiority is evident not only in typical fine-grained datasets (CUB, Stanford Cars, and Stanford Dogs) but also in more challenging fine-grained datasets (meta-iNat and tiered meta-iNat). Furthermore, compared to AGPF based on multi-scale feature aggregation, our method consistently exhibits performance improvements across all experimental settings. This not only indicates the importance of filtering out background and other trivial noise while introducing mid-level features, but also validates the effectiveness of the SGNF module. Additionally, under the 1-shot setting with only one support sample, our method demonstrates a more significant leading advantage, indirectly proving the effectiveness of BTG-Net in preventing overfitting. It also confirms that, by introducing the TG prompt, based on cross-task general knowledge learned, the model can be well generalized to novel tasks with very limited task-specific support samples. Finally, BTG-Net exhibits excellent performance on both Conv-4 and ResNet-12, showing nonsignificant overfitting with an increase in feature extractor parameters.

### 4.4 Analysis

To further demonstrate the effectiveness of the two modules and the rationality of the detailed designs within the modules, we conduct additional experiments and analyses.

*4.4.1 Ablation study of submodules.* In order to demonstrate the effectiveness of the two modules proposed, we conduct the module-wise ablation study as shown in Table 5. The experimental results

**Table 6: Results of different layer integration ways in the SGNF module on the CUB and Stanford-Dogs datasets, where $l$, $s$, and $t$ represent the outputs of the last layer, second-to-last layer, and third-to-last layer of the network respectively, Concat($\cdot$) refers to the concatenation operation, and SGNF($a, b$) denotes that the SGNF preserves layers $a$ and $b$.**

| Model | CUB | | Stanford-Dogs | |
|---|---|---|---|---|
| | 1-shot | 5-shot | 1-shot | 5-shot |
| Baseline | 68.03 | 81.92 | 55.85 | 71.35 |
| Concat($l, s, t$) | 70.44 | 83.98 | 57.36 | 73.09 |
| SGNF($l, s$) | 76.66 | 87.91 | 65.51 | 79.55 |
| SGNF($l, t$) | 77.20 | 88.12 | 64.67 | 79.04 |
| SGNF($l, s, t$) | **79.19** | **89.71** | **67.33** | **81.08** |

**Table 7: Experimental results of removing the TG Prompt in the GKPM on the CUB and Stanford-Dogs datasets.**

| TG Prompt | CUB | | Stanford-Dogs | |
|---|---|---|---|---|
| | 1-shot | 5-shot | 1-shot | 5-shot |
| w/o | 78.02 | 89.07 | 65.56 | 79.86 |
| w/ | **79.19** | **89.71** | **67.33** | **81.08** |

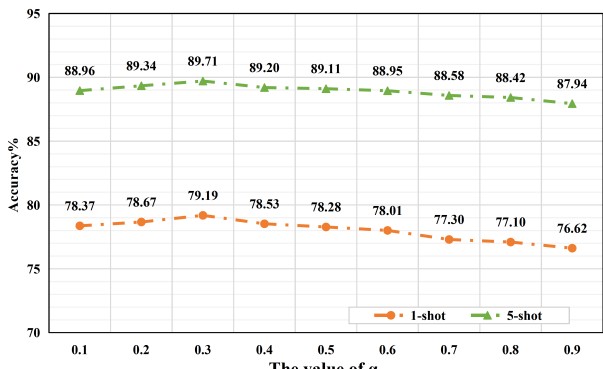

**Figure 3: Classification performance of the BTG-Net with different values of $\alpha$, using Conv-4 backbone on the CUB dataset.**

demonstrate that both SGNK and GKPM enhance baseline performance. When the two modules are combined, the performance sees a significant further improvement, validating the effectiveness of the proposed method in addressing FS-FGIC tasks from both FGIC and FSL perspectives.

*4.4.2 Effectiveness analysis of SGNK.* To demonstrate that the SGNK module is an ideal framework for leveraging the feature learning capability of the network in a multi-layer strategy, we conducted additional experiments, and the results are shown in Table 6. The results show that the concatenation operation of multiple layers inevitably introduces noise, leading to overfitting issues under the FSL setting, thus, the performance improvement is not significant.

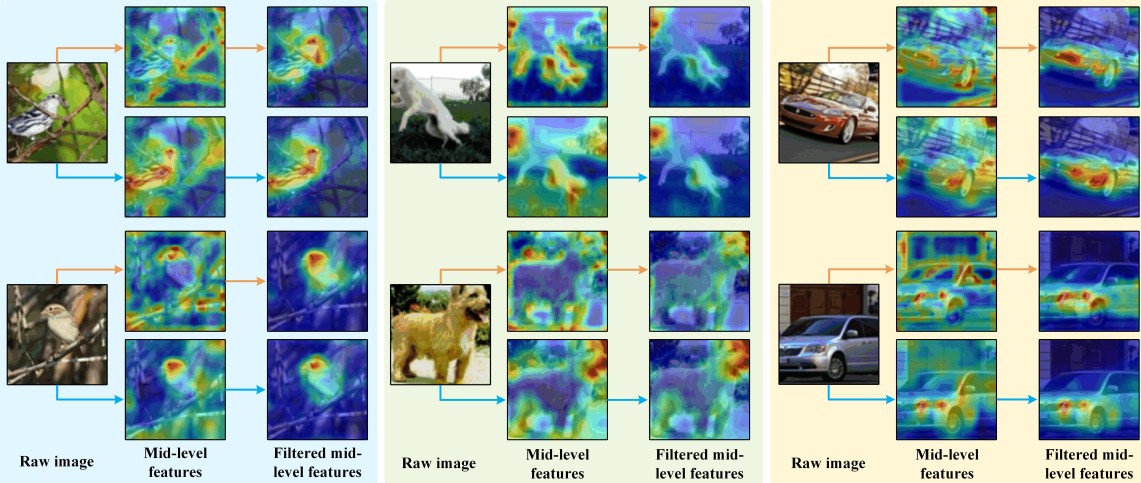

**Figure 4: The visualization results of mid-level features and the filtered mid-level features corresponding to the raw image on the CUB, Stanford-Dogs and Stanford-Cars datasets.**

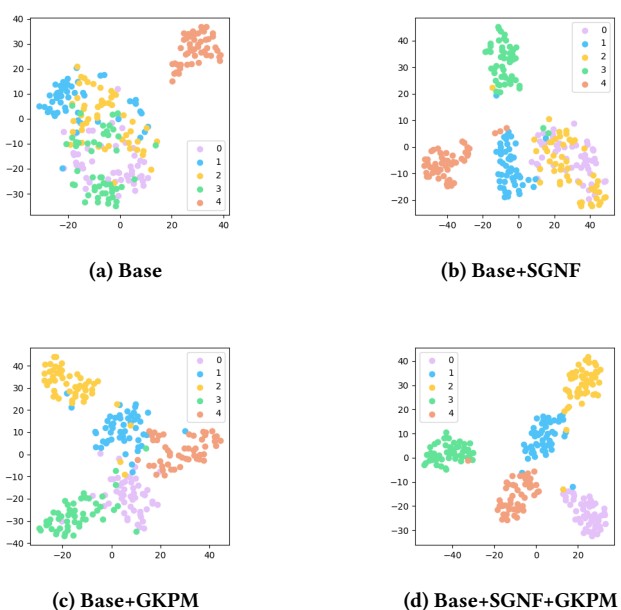

(a) Base

(b) Base+SGNF

(c) Base+GKPM

(d) Base+SGNF+GKPM

**Figure 5: 5-way t-SNE visualization results of the embedding space on the CUB dataset, where different colors represent different classes. Randomly selecting five novel classes from the test set, and each class contains 50 query samples.**

Therefore, it is essential to filter out background noise while introducing mid-level features, and the SGNK module designed in this paper proves to be the most effective structure.

*4.4.3 Effectiveness analysis of TG Prompt in the GKPM.* We conducted additional experiments with (w/) and without (w/o) TG Prompt, as shown in Table 7. The results indicate that TG Prompt contributes to performance improvement, further confirming that

learning cross-task general knowledge is effective for FS-FGIC, and can enhance the generalization ability of the model on novel classes.

*4.4.4 The influence of $\alpha$.* To better explore the impact of balancing hyperparameter $\alpha$ on model performance, we conducted additional experiments as shown in Figure 3. The experimental results indicate that setting the value of $\alpha$ to 0.3 leads to the highest classification accuracy. This indicates the necessity of hyperparameter $\alpha$.

## 4.5 Visualization

To fully demonstrate the effectiveness of the SGNF module, we present additional visual results as shown in Figure 4. Based on the results, it can be observed that the proposed SGNF module effectively eliminates background noise in the mid-level features.

To better understand the effectiveness of each module from the perspective of feature mapping results, we also show the t-SNE visualization results in Figure 5. It can be observed that results generated by only a baseline model (Figure 5a) are unsatisfactory, with blurry class boundaries and many points of different classes intermingled. According to Figure 5d, it can be observed that, after combining SGNF and GKPM, each class forms a compact cluster and the boundaries between different clusters are obvious. This further demonstrates the effectiveness of SGNF and GKPM, then verifies that our proposed model has strong generalization ability and can learn optimized fine-grained features.

## 5 Conclusion

In this paper, we propose a novel Bi-directional Task-Guided Network (BTG-Net) for FS-FGIC. The BTG-Net consists of two modules. The Semantic-Guided Noise Filtering (SGNF) module effectively filters out noise such as background in mid-level features, and the General Knowledge Prompt Modeling (GKPM) module ensures robust generalization performance on novel classes by introducing cross-task general knowledge. Our proposed method achieves comprehensive leadership in performance on five benchmark datasets.

# Acknowledgments

This work was supported in part by the National Natural Science Foundation of China under Grant 62202272, 62172256, 62202278, in part by Natural Science Foundation of Shandong Province under Grant ZR2019ZD06, ZR2020QF036, ZR2021ZD15, in part by the Young Scholars Program of Shandong University, and in part by the Major Program of the National Natural Science Foundation of China under Grant 61991411.

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
