# OpenReview forum: "Bi-directional Task-Guided Network for Few-Shot Fine-Grained Image Classification"
_acmmm.org/ACMMM/2024/Conference — MM2024 Poster_

### Official Review · Reviewer_pwKF · 2024-05-24

**Rating:** 3
**Confidence:** 4

**Summary:**

This paper introduces a Bi-directional Task-Guided Network (BTG-Net) to tackle Few-Shot Fine-Grained Image Classification (FS-FGIC) task. Extensive experiment results on five few-shot fine-grained benchmark datasets demonstrate that the proposed network outperforms state-of-the-art methods.

**Strengths:**

1. The proposed methods and experimental results are well presented, and the manuscript is well organized overall.
2. The performance gain is impressive when compared to state-of-the-art methods.

**Limitations:**

1. The author should consider whether "Bi-directional" in the title is appropriate.

2. The author should discuss the performance impact of the feature compression block.

3. Figure 2 should be readjusted to avoid ambiguity.

4. The formulas and symbols in Chapter 3 need to be reorganized. For example:
a) There is an error in formula (3).
b) The dimension of $F^{S}$ should be added to $R^{N \times K \times C \times H \times W}$, otherwise, formula (10) and $Avg$ in Figure 2 are not clear.
c) Formula (12) and formula (13) are misleading because $W_{qkv}$ is unclear.
d) Formula (2) and formula (3) are inconsistent with the description in Figure 2.
e) Many formulas and terms, such as $B(\cdot)$, need more explanations.

5. In ResNet-12, how is the middle layer set up?

6. There are many typos, such as BS-Net on line 216 and missing punctuation on line 906.

7. Consider adding the latest FS-FGIC articles and results for comparison, such as:

-  J. Wu et al., Bi-Directional Ensemble Feature Reconstruction Network for Few-Shot Fine-Grained Classification. TPAMI, 2024.

-  Z.-X. Ma et al., Cross-Layer and Cross-Sample Feature Optimization Network for Few-Shot Fine-Grained Image Classification. AAAI, 2024.

**Suitability:**

2

---

### Official Review · Reviewer_yWya · 2024-05-24

**Rating:** 5
**Confidence:** 3

**Summary:**

This paper focus on few-shot fine-grained classification task and designs two modules from the perspectives of few-shot task and fine-grained classification task respectively. This paper proposes a Semantic-Guided Noise Filtering (SGNF) module to reduce the impact of noise on model performance during feature fusion. This paper also proposes a General Knowledge Prompt Modeling (GKPM) module to retain cross-task knowledge to improve model performance on novel classes. The proposed model achieves SOTA results on five few-shot fine-grained benchmark datasets.

**Strengths:**

This paper proposes two novel modules to simultaneously enhance model performance on few-shot and fine-grained classification task.
In the SGNF module, using high-level features to filter out noise from mid-level features can effectively retain useful fine-grained information.
In the GKPM module, a trainable prompt is used to extract general knowledge from episode training process, which improves model’s generalization capability on novel classes.
Alation studies demonstrate the effectiveness of two modules and the whole model achieves SOTA results on five benchmarks.

**Limitations:**

The trainable prompt may have learned dataset-aware information, which is invalid for the new test dataset.
This paper did not conduct cross-dataset test experiments.
The iteration term $U$ in formulation 18 has not been explained.

**Suitability:**

3

---

### Official Review · Reviewer_YrD1 · 2024-05-25

**Rating:** 5
**Confidence:** 4

**Summary:**

This proposes a novel Bi-directional Task-Guided Network (BTG-Net) for Few-shot Fine-Grained Image Classification. Especially, the Semantic-Guided Noise Filtering (SGNF) module to filter noise on mid-level features rich in detailed information. Moreover, the General Knowledge Prompt Modeling (GKPM) module is proposed to retain the cross-task general knowledge by utilizing the prompting mechanism, thereby enhancing the model’s generalization performance on novel classes. The extensive experiments on five few-shot fine-grained benchmark datasets demonstrate the effectively of the proposed method.

**Strengths:**

1、The motivation is reasonable.

2、The writing is good, and the proposed method is clear to understand.

3、Extensive evaluation verifies the effectiveness of the proposed method.

**Limitations:**

1、Thanks to the high generalization of the Visual-Language Models (VLMs), the CLIP with the prompt-tuning methods shows impressive performance on zero-shot learning and few-shot learning. Therefore, what is the advantage of the proposed method compared to CLIP with prompt tuning for the few-shot fine-grained image classification?

2、It lacks the comparison on the backbone of ResNet-18. Moreover, why the comparison of meta-iNat and tiered meta-iNat is only using the Conv-4 backbone and without considering the ResNet12 similar to the previous comparison?

3、What is the core motivation of the General Knowledge Prompt Modeling, and how it can solve the problem in Few-shot Fine-Grained Image Classification?

**Suitability:**

3

---

### Official Review · Reviewer_tjwK · 2024-05-25

**Rating:** 3
**Confidence:** 3

**Summary:**

The paper addresses the Few-Shot Fine-Grained Image Classification (FS-FGIC) problem by proposing a novel Bi-directional Task-Guided Network (BTG-Net). This network aims to enhance performance by tackling the cross-task challenges in FS-FGIC. From the Fine-Grained Image Classification perspective, it introduces a Semantic-Guided Noise Filtering (SGNF) module to filter noise in mid-level features, which are rich for the task. From the Few-Shot Learning perspective, it employs a General Knowledge Prompt Modeling (GKPM) module to retain cross-task general knowledge, leveraging the prompting mechanism for better generalization to novel classes. Extensive experiments on five benchmark datasets demonstrate that BTG-Net outperforms state-of-the-art methods.

**Strengths:**

- **Clarity:** the paper and proposed method are easy to follow and easy to grasp.
- **Extensiveness of experiments:** the amount of datasets and baseline method prove the strength of the proposed method. Nevertheless, refer to the first limitation for a doubt about baseline evaluation.

**Limitations:**

- **Reported baseline results:** The reported numbers for BiFRN [44] in the quantitative evaluation do not match those in the original paper, in some of the cases they are off by more than 3%, and the paper does not state whether the results are a re-implementation by the authors. Given that some of the original results are higher than those obtained by the proposed method, a clarification on the matter is crucial.
- **Role of $\alpha$:** the paper does not provide an explanation of the role of the two merged branches and the interpolation through the use of $\alpha$. Since both of the branches are convolutional and act on the same input, how are they differentiated?

**Suitability:**

2

---

### Meta-Review · Area_Chair_hfJ1 · 2024-07-02

**Recommendation:** Accept (Poster)
**Confidence:** 4

**Metareview:**

The paper presents an approach to few-shot fine-grained image classification.
The approach is agreed by most reviewers to be new. The experiments and results are considered to be strong by all reviewers.
Authors rebuttal was deemed as effective by all except one reviewer. The lingering issues raised by the lone reviewer who voted for weak reject seem to be minor and fixable in the final paper. Hence this paper should be in the accepted batch.